# Metatranscriptome Analysis of Nasopharyngeal Swabs across the Varying Severity of COVID-19 Disease Demonstrated Unprecedented Species Diversity

**DOI:** 10.3390/microorganisms11071804

**Published:** 2023-07-14

**Authors:** Michaela Hyblova, Dominik Hadzega, Klaudia Babisova, Patrik Krumpolec, Andrej Gnip, Peter Sabaka, Stefan Lassan, Gabriel Minarik

**Affiliations:** 1Medirex Group Academy, 949 05 Nitra, Slovakiapatrik.krumpolec@medirexgroupacademy.sk (P.K.); gabriel.minarik@medirex.sk (G.M.); 2Department of Infectology and Geographical Medicine, Faculty of Medicine, Comenius University in Bratislava, 814 99 Bratislava, Slovakia; 3Department of Pneumology and Ftizeology I, University Hospital in Bratislava, 831 01 Bratislava, Slovakia

**Keywords:** metatranscriptome, microbiome, *SARS-CoV-2*, virome

## Abstract

The recent global emergence of the *SARS-CoV-2* pandemic has accelerated research in several areas of science whose valuable outputs and findings can help to address future health challenges in the event of emerging infectious agents. We conducted a comprehensive shotgun analysis targeting multiple aspects to compare differences in bacterial spectrum and viral presence through culture-independent RNA sequencing. We conducted a comparative analysis of the microbiome between healthy individuals and those with varying degrees of COVID-19 severity, including a total of 151 participants. Our findings revealed a noteworthy increase in microbial species diversity among patients with COVID-19, irrespective of disease severity. Specifically, our analysis revealed a significant difference in the abundance of bacterial phyla between healthy individuals and those infected with COVID-19. We found that *Actinobacteria*, among other bacterial *phyla*, showed a notably higher abundance in healthy individuals compared to infected individuals. Conversely, Bacteroides showed a lower abundance in the latter group. Infected people, regardless of severity and symptoms, have the same proportional representation of *Firmicutes*, *Proteobacteria*, *Actinobacteria*, *Bacteroidetes*, and *Fusobacteriales*. In addition to *SARS-CoV-2* and numerous phage groups, we identified sequences of clinically significant viruses such as *Human Herpes Virus 1*, *Human Mastadenovirus D*, and *Rhinovirus A* in several samples. Analyses were performed retrospectively, therefore, in the case of *SARS-CoV-2* various WHO variants such as *Alpha* (B.1.1.7), *Delta* (B.1.617.2), *Omicron* (B.1.1.529), and *20C* strains are represented. Additionally, the presence of specific virus strains has a certain effect on the distribution of individual microbial taxa.

## 1. Introduction

At every breath, we inhale an airborne cocktail containing thousands of microbial particles. The upper respiratory tract is a major gateway for a large number of viruses during infection, not just respiratory ones. At the same time, the mucosa of the nasal cavity is colonized by a wide spectrum of microorganisms whose specific importance is not well elucidated. Although this massive burden may sound frightening, it is undoubtedly essential in building the immunity of individual hosts. A healthy upper respiratory tract microbiota works in tandem with its host, mainly colonizing the anterior nares and nasopharynx to provide an innate barrier that defends against pathogens and modulates the immune responses that occur when exposed to external triggers such as smoke, dust, allergens, chemicals, temperature changes, and microorganisms [1]. Under the influence of the recent COVID-19 pandemic, interest in how viral infection affects/modifies the overall microbiome of the airways, particularly the well-accessed upper respiratory tract, has gained prominence. Currently, many scientific reports explore microbial diversity in the context of COVID-19 disease severity [2,3,4] unfortunately often with controversial outcomes. Besides COVID-19, many studies have demonstrated that unique microbial communities have complex interactions with the host to maintain balance with the host immune system [5,6].

The nasal microbiome of healthy humans is composed primarily of *Actinobacteria*, *Bacteroidetes*, *Firmicutes*, and *Proteobacteria phyla*, with a predominance of representatives of the genera *Bifidobacterium*, *Corynebacterium*, *Staphylococcus*, *Streptococcus*, *Dolosigranulum*, and *Moraxella* [7,8]. *Archaea* are not typically found in the upper respiratory tract of humans, as they are not known to be part of the normal microbiota. However, there have been some studies that have identified the presence of *archaea* (*Methanoarchaea*) in the nasal cavity and other parts of the body [9].

Among the bacteria, *Bacteroidetes* and *Firmicutes* predominate in most of the anatomical regions studied, including the upper respiratory tract (URT), and are the most investigated components of the human microbiota [10]. 

A somewhat less well-known fact is that humans are colonized by a remarkable number of DNA and RNA viruses that are referred to as the virome, capable of causing acute, persistent, or latent infection, and retroviral elements integrated to host chromosomes. The human virome consists of bacteriophages (phages) that infect bacteria, archaeal viruses, eukaryotic viruses that infect human cells, and viruses transient in food [11,12]. Among the viral genomes identified, *Picornaviruses*, *Anelloviruses*, and bacteriophages of the family *Siphoviridae* were the most prevalent in the upper respiratory tract [13]. The human virome comprehends commensals and opportunistic pathogens. The balance between being a commensal or becoming a pathogen is determined by different factors of the viral community itself and the host, such as genetic factors and immune status [14]. It has been well-documented that resident viruses can cause exacerbation of chronic pulmonary diseases, cystic fibrosis, and asthma. Moreover, they may contribute to the pathogenesis of community-acquired respiratory virus infections [15].

Generally, there are two major approaches to detecting microbial spectra: 16S rRNA amplicon sequencing and metagenomic shotgun sequencing. The former is attractive at lower cost, is preferred in environments with lower microbial diversity, and identifies mainly dominant microorganisms [16]. The latter is characterized by a more robust resolution and covers the metagenome community including the human genome, viruses, and fungi, which cannot be captured by 16S amplicon sequencing [17]. In our approach, we replace the analysis of metagenome with the metatranscriptome to cover also RNA viruses, which include *SARS-CoV-2*. However, we actually did not sequence the total RNA but depleted the dominating human 18S rRNA in the initial steps of library preparation to increase the relative abundance of other RNA belonging to humans and other microbial agents. In general, 18S rRNA is estimated to make up around 80–90% of the total RNA in a typical mammalian cell [18].

## 2. Material and Methods

### 2.1. Patients and Samples

The study was conducted in accordance with the approval of the Ethics Committee of the Bratislava Self-Governing Region under the identification number 03228/2021/HF, dated 12 January 2021. All patients signed informed consent and received questionnaires with relevant information regarding their medical history and health status in relation to COVID-19. All 151 patients were from Slovakia and were divided into four groups: asymptomatic (n = 24), patients with mild/moderate (n = 25), severe COVID-19 (n = 30), and a negative control group (n = 72). The group of patients with severe COVID-19 was recruited from two university hospitals in Bratislava: Ruzinov and Kramare (Slovakia).

The timeframe of sampling in hospitalized patients was different in the two hospitals. At Ruzinov Hospital, patients were enrolled in the study at approximately day 7–8 (median 7) after admission to the hospital, while at Kramare Hospital it was day 3–4 (median 3.5). The treatment regimen was also different in the two hospitals. In Ruzinov Hospital, 100% of patients received antibiotic treatment and corticotherapy (corticosteroid treatment) (18/18), more than 60% of patients received immunomodulatory therapy (20/28), and no patient received antiviral treatment. In the second hospital, on the other hand, less than 20% received antibiotic therapy (2/12) in combination with corticosteroids (8/10), immunomodulatory and antiviral therapy (5/10). Mild and asymptomatic patients were recruited from mobile collection sites operated by Medirex, the second largest provider of laboratory diagnostics in Slovakia. Negative controls were recruited among Medirex employees and their family members. A negative RT-PCR test and asymptomatic status were prerequisites. Randomly captured RT-PCR-positive but asymptomatic patients were reassigned to the asymptomatic group. Age and sex characteristics are summarized in Table 1. Samples were analyzed in the order in which they arrived, in proportional representation, and no preference was given to age, sex, or disease severity when selecting for sequencing. Nasopharyngeal swabs were collected from March 2021 to October 2022 in viRNAtrap transport medium (GeneSpector Innovations, Prague, Czech Republic) and stored at 4 °C in a refrigerator until processed.

### 2.2. RNA Isolation, RT-PCR and Library Preparation

RNA was isolated retrospectively from a nasopharyngeal swab-originated collection medium representing samples from patients with positive findings for COVID-19, regardless of disease severity, without selection based on viral load (high or low Ct values) and from COVID-negative controls using the Cytiva^®^ Sera-XtractaTM virus/Pathogen Kit (Global Life Sciences Solutions Operations, Little Chalfont, UK) on a KingFisher Flex benchtop automated machine (ThermoFisher Scientific, Waltham, MA, USA). RNA quantity was measured fluorometrically using QubitTM RNA High sensitivity (Invitrogen, Eugene, OR, USA). RNA isolates were stored at −80 °C until processed. Genomic libraries were prepared using Kappa HyperPrep with RiboErase Kit (Kapa Biosystems, Salt River Cape Town, South Africa) with depletion of eukaryotic RNA for 18S rRNA according to the manufacturer’s recommendations. Illumina’s TruSeq CD dual adapters were used for sample indexing. The resulting libraries were quantified by fluorometric analysis using the QubitTM dsDNA HS Assay kit (Invitrogen, USA) and fragment analysis using the High Sensitivity DNA reagent kit (Agilent Technologies, Waldbronn, Germany).

All RNAs underwent RT-PCR to confirm or exclude *SARS-CoV-2* positivity using the COVID-19 Real-Time Multiplex RT-PCR Kit (Labsystems Diagnostics, Vantaa, Finland) and the ABI QuantStudio 6 Real-Time PCR System RT-qPCR platform (ThermoFisher Scientific, Waltham, MA, USA) using the original manufacturer’s protocols. A Ct value < 40 was required to evaluate a sample as positive.

### 2.3. Sequencing and Bioinformatic Analysis

Paired-end sequencing (2 × 75 and 2 × 100) was performed on NextSeq500/550 and NextSeq2000 (Illumina, San Diego, CA, USA) platforms, respectively. Paired-end sequencing data were subjected to quality control by FastQC v0.11.9 [19] and reads were processed by Trimmomatic v0.39 (CROP:96 HEADCROP:10 LEADING:22 TRAILING:22 SLIDINGWINDOW:4:22 MINLEN:25 and our own set of adapter sequences was used in the ILLUMINACLIP step) [20]. First, reads were mapped to the human reference hg38 (GRCh38) by the BWA-MEM algorithm (from bwa v0.7.17) [21]. Unmapped reads, extracted using *samtools view* from samtools v1.6 [22] and *Picard SamToFastq* from Picard v2.27.4 [23], longer than 50 bp were further processed for microbiome identification. To identify and quantify bacterial species, we used Kraken2 v2.1.2 [24], as we previously observed to be well-functional for this purpose [25]. Kraken2 was run in paired-end mode with the minimum-base-quality set on 20, while the minimum-hit-groups parameter was set on 2 and a standard database was used (built with a standard flag, installed on 2 August 2022). Flag-use-mpa-style was used to produce output, which can be useful in downstream analysis and is also more easily readable.

For the *SARS-CoV-2* virus, we applied de novo assembly algorithm coronaspades.py from Spades v3.15.5 [26]. *SARS-CoV-2* lineage/clade assignment was carried out as proposed in Galaxy tutorial for a pipeline constructed by Maier and Batut [27].

### 2.4. Statistical Analysis

To statistically analyze the results of Kraken2, mpa-style outputs were first brought together by combine_mpa.py from KrakenTools v1.2 [28] counts were normalized by division by the total read pairs number in the given sample (after Trimmomatic step) and subsequently multiplied by the average read pairs number. The difference in the number of bacterial transcripts detected between groups was statistically analyzed by LefSE tool (Galaxy Version 1.0) [29].

The normalized data were further analyzed using the R programming language. Firstly, Bray–Curtis dissimilarity distance matrices were computed for all pairwise patient group comparisons. PERMANOVA was then performed to compare the distribution of bacterial genera and species between each pair of groups. Mann–Whitney U test (Wilcoxon test) was used to compare the counts of reads matching sequences from the Kraken2 database between each pair of groups. This test was also applied to compare Ct values between each pair of positive groups, thus excluding negative controls. To account for multiple comparisons, Bonferroni correction was used to set a threshold of significance for *p*-values.

To examine the abundance data in relation to various covariates and factors such as age, sex, severity, ATB treatment, and SARS CoV-2 variant, we conducted additional statistical analysis using the R package Maaslin2 [30]. The Maaslin2 function was applied to the pre-normalized data with specific modifications to the options: a minimum abundance threshold of 100 (min_abundance = 100), no standardization (standardize = FALSE), no additional normalization (normalization = “NONE”), and no transformation (transform = “NONE”). Consequently, a taxon had to exhibit an abundance value of at least 100 in at least one sample to be considered for further analysis.

By default, a linear model was employed as the analysis method (analysis_method = “LM”), and the Benjamini–Hochberg correction method (correction = “BH”) was utilized. The analysis encompassed all available samples, incorporating the following parameters: severity, SARS-CoV-2 WHO variant, age (as continuous data), age category (as categorical data), sex, and ATB treatment. Additionally, we examined specific categories individually, including severe patients alone (to assess the impact of antibiotic treatment) and negative controls alone (to explore potential effects of age and sex on microbial transcript quantities).

## 3. Results

### 3.1. Prokaryotic Microbiome

By metatranscriptome shotgun sequencing, we analyzed 151 samples categorized into four main groups: asymptomatic (24, A), mild (25, M), severely affected (30, S) and control group (72, NC). The mean number of read pairs per sample after trimming was 45.3 M (27.7–133 M). Eukaryotic 18S rRNA from nasopharyngeal swabs was previously removed during the DNA library preparation (detailed in Section 2).

The microbiome fraction or simply the number of reads matching sequences from the standard Kraken2 database, distribution, and representation of bacterial taxa were different across all four groups. The relative abundance of bacteria according to mapped sequence reads was significantly higher in asymptomatic and mild patients. *Firmicutes*, *Bacteroidetes*, *Proteobacteria*, *Actinobacteria*, and *Fusobacteriales* were the most abundant *phyla* among the positives. On the contrary, in the control group, *Actinobacteria*, *Proteobacteria*, and *Firmicutes* were the most abundant group while *Bacteroides* and *Fusobacteriales* had only a negligible proportion (Figure 1A–D; Figure 2A,B). Using a threshold of 1000 reads in at least one sample, we identified a total of 944 species, 531 genera, 218 families, 110 orders, 28 classes, and 25 phyla. The highest number of species was in the severe (725), asymptomatic (692), mild (574), and less than tenfold lower numbers in the negative control group (58). It is important to mention that 20 out of 30 severe patients from two hospitals, including 100% of patients from Ruzinov Hospital (18/18) and 16.7% of patients (2/12) from Kramare Hospital, were treated with antibiotics. Antibiotic treatment (ATB) resulted in a reduction of the total bacterial count, but the ratio of individual bacterial taxa was nevertheless maintained (Figure 2C). Although we observed a decrease in the abundance of major bacterial taxa in severe antibiotic users and non-users, this difference was not statistically significant (Figure 2D). Apart from researching bacterial content, we compared relative counts of taxa transcripts between different disease symptoms and different *SARS-CoV-2* variants (*alpha*, *delta*, *omicron*) by LefSE tool. Here, we are showing differentially represented genera under the LDA threshold, further filtered by median values (Figure 2E).

We further profiled the microbial composition at the genus and species level in healthy controls and various COVID-19 groups. In the nasopharyngeal swab samples, there is a striking enrichment of *Streptococcus*, *Prevotella*, and *Veillonella*, the most abundant genera in the mild and asymptomatic groups (Figure 3A). *Stenotrophomonas*, *Staphylococcus*, and *Corynebacterium* are dominating genera in the severe group. Surprisingly again, *Stenotrophomonas*, *Mycobacterium*, and *Pseudomonas* are the most abundant in healthy participants. At the species level, *Stenotrophomonas maltophilia*, *Cutibacterium acnes*, and *Halomonas* sp. *JS92-SW72* were enriched in severe patients. *Veillonella atypica* and *Prevotella melaninogenica* were most abundant in the mild and asymptomatic groups, and the total numbers of other species of the genus *Prevotella* (*P. jejuni*, *P. histicola*, *P. intermedia*, *P. oris*) and *Streptococcus* (*S. parasanguinis*, *S. mitis*, *S.* sp. *LPB0220*) were very similar (Figure 3B). The most frequently present species in the healthy control group as well as in the severe group were *Stenotrophomonas maltophilia*, *Halomonas* sp. *JS92-SW72* and *Mycobacterium canettii*. Of the clinically relevant opportunistic bacterial species, e.g., *Klebsiella pneumoniae*, *Staphylococcus aureus*, *Streptococcus pneumoniae* and *Haemophilus parainfluenzae* were present in all four, although they did not significantly dominate in any of the groups and the differences between the groups were not statistically significant; rather, their prevalence was similar in the pairs of severe and negative control versus mild and asymptomatic. Of particular interest was the presence of bacterial species in all groups that are not entirely typical of the human upper respiratory tract microbiome. Of both Gram-negative *coccobacilli*, *Moraxella osloensis* is rarely isolated from clinical specimens and *Pasteurella multocida* is part of the normal microbiota in the nasopharynx of many wild and domestic animals, including cats and dogs.

By reducing the threshold from 1000 reads to 100, we were able to identify sequences of microorganisms that are not typical of the nares and nasopharynx microbiome, e.g., *Archaea* (originally *Archebacteria*), a distinct group separate from bacteria that commonly inhabit extreme biotypes. We identified two phyla *Candidatus Micrarchaeota* and *Euryarchaeota*. The most abundant species identified was *Natrinema salinisoli*.

### 3.2. Virome Analyses

Among the 79 RT-PCR-positive samples, 51 complete *SARS-CoV-2* genomes and 19 partially complete *SARS-CoV-2* genomes (multiple scaffolds) were assembled by metatranscriptome shotgun sequencing, accounting for 62% (51/79). Additionally, we did not identify *SARS-CoV-2* sequences in any of the control group samples (*SARS-CoV-2* RT-PCR negative). In the asymptomatic, mild, and severe group, we identified *SARS-CoV-2* clades: *Alpha* (B.1.1.7)—clade 20I, *Delta* (AY.4; AY.43; AY.43.9; AY.122; AY.9.2)—clade 21I and in one case 21J, *Omicron* (BA.1.1; BA.2.9; BA.2; BA.2.67; BA.5)—clade 21L and in one case 22B, and one sample clade 20C. The *Alpha* and *Delta* variants were predominant in the severe COVID group; however, 30% of analyses failed to assemble the genome due to the insufficient read counts aligned to the *SARS-CoV-2* reference. *Alpha* was dominant in the mild group, while *Omicron* and *Delta* were dominant in an asymptomatic group with an undetermined *SARS-CoV-2* clade in 24% and 33% with median Ct values of 33.3 and 33.8, respectively.

In addition to *SARS-CoV-2*, we identified sequences of other human RNA viruses, e.g., from the family *Picornaviridae* (*Rhinovirus A*). As expected, among the DNA viruses, these were mainly phages from *Pedoviridae*, *Rountreeviridae*, and *Siphoviridae*. In a few individual samples, we identified sequences of *Human Herpes Virus 1*, *HSV-1 (Alphaherpesviridae)* and *Human Mastadenovirus D (Adenoviridae)*.

In terms of age, the differences were mainly between severe COVID patients from the hospital (median 68) and the other groups (M-37, A-42, N-37) (Table 1). But the distribution of taxa in age-similar patients was similar in the microbiome representation in all positive participants.

We performed additional linear statistical analyses (Maaslin2) to correlate the abundance data along with age, sex, and antibiotic treatment. When analyzing age as a continuous variable, we did not observe any discernible effect on the bacterial abundance. However, when considering age as a categorical variable, we found significant effects on six specific taxa. Expanding our analysis to include all samples, we discovered variations in the abundance of 231 taxa that correlated with age (Appendix A). When we focused exclusively on severe patient samples, which tended to include older individuals, we observed a distinct and pronounced influence of age. Moreover, 114 taxa showed significant associations with antibiotic (ATB) treatment in severe patients (Appendix A).

## 4. Discussion

Our study compares the microbiome including the virome of the nasopharynx among four groups of participants based on the severity of the viral respiratory disease: negative (no signs of illness, RT-PCR negative), mild (benign symptoms, not requiring hospitalization), asymptomatic (no signs of illness, RT-PCR positive), and severely ill COVID-19 patients (breathing difficulties, pneumonia, hospitalized) using metatranscriptome shotgun sequencing. Samples were collected over an extended period of time at mobile sampling centers (NC, asymptomatic, mild) and at two hospitals (severe). In the text, for simplification, we use an abundance of bacteria or viruses, although, in reality, we are still referring to transcripts, which, however, express the representation of the active part of the microbiome. In addition, filtering for contaminating reads was particularly important in the context of host-associated metatranscriptomic datasets as sequences of host origin can represent a significant proportion of reads in a sample. Therefore, we already removed human 18S rRNA via depletion during library preparation. For bioinformatic analysis, we analyzed reads that did not map to the human reference based on length and quality, and further analyzed using the Kraken2 tool (Section 2).

There are additional concerns in microbiome research that need to be addressed, specifically regarding potential contamination from the environment or cross-contamination from other samples. However, systematic requirements in this regard are still lacking. Initially, RNA depletion and bioinformatics algorithms may not be sufficient to overcome this issue. The inclusion of sample and extraction blanks, along with non-sample controls in the PCR step, could be valuable in reducing the likelihood of contamination.

The abundance of bacteria is significantly higher in the mild and asymptomatic groups while only a small proportion is present in the severe and negative. The low abundance of bacteria in severe patients is easily explainable by the intake of antibiotics. The use of antibiotics without a clear indication of bacterial superinfection is controversial; however, it appears to have been common practice in some healthcare centers regardless of the risks of antimicrobial resistance. The side effects of corticosteroids and immunomodulatory therapies on the microbiome of the upper respiratory tract are complex and can vary depending on multiple factors such as dosage, duration, and individual factors; however, they can alter the microbial diversity in both ways: increase or decrease [31]. Few publications report that the bacterial load increased as COVID-19 severity increased (although differences between groups were not statistically significant), and the abundance of the microbiome increases with the severity of the disease [32,33]. According to the severity of the disease, *Firmicutes* are most frequent in asymptomatic, mild, and severe patients, whereas they are significantly less abundant and proportionally represented in healthy controls (Figure 1A–D). We observed a significant decrease in the abundance, but not in the diversity of bacterial taxa and in the representation of major taxa in severe patients. Although even in a small group of 30 patients, we divided them into antibiotic-treated and non-antibiotic-treated patients, apart from the apparent decrease in abundance, we did not observe a change in the relative abundance of the different bacterial phyla (Figure 2C,D).

The division into two groups of COVID-positive mild and asymptomatic was motivated by the objective to investigate whether there is any difference between the microbiome of people with symptoms of respiratory disease such as rhinitis, fever, cough and people who, although infected (RT-PCR positive), do not show any symptoms. Our results indicate that both the abundance of bacteria and the proportions of representation of each taxon are very similar in both groups. Although the median Ct values in mild (24.89) and asymptomatic (28.56) appeared to be rather different, they were not statistically significant (Wilcoxon test, *p* = 0.017), the size of both groups (25 vs. 24), and age distribution (37 vs. 42) were otherwise similar. We did not observe any significant differences at the genus and species level, as confirmed by the statistical analysis (PERMANOVA; *p* = 0.053 and 0.024, calculated cut-off *p*-value was 0.0023). The most enriched genera in both groups were *Streptococcus*, *Veillonella* (both *phyla Firmicutes*), and *Prevotella* (*phyla Bacteroidetes*) with *Veillonella atypica* and *Prevotella melaninogenica* being the most abundant species (bacterial species and genus (Figure 3B)). This observation is consistent with findings from similar studies of the microbiome of upper respiratory airways [34,35,36]. *P. melaninogenica* is a Gram-negative, obligate anaerobic coccobacillus that can act as an opportunistic pathogen, and there are conflicting hypotheses for its effect on the respiratory tract. For example, *P. melaninogenica* has been found to be a “beneficial” member of the airway microbiome because it enhances protection against bacterial pneumonia caused mainly by *Streptococcus pneumoniae* (*S. pneumoniae*). Further, *Prevotella melaninogenica* was ranked among the most distinguishing bacterial species separating patients with pneumonia caused by *S. pneumoniae* (with less *P. melaninogenica*) from healthy controls (with more *P. melaninogenica*) [37]. Interestingly, the severe group with acquired pneumonia of unknown bacterial origin had significantly less *P. melaninogenica* (Figure 3B).

We analyzed 72 negative healthy controls, all were additionally verified/tested by RT-PCR (genes *E*, *ORF1ab*, *N*) and confirmed negative. The highest bacterial abundance based on the number of mapped reads and transcripts was in the asymptomatic and mild groups. This is not that surprising since any viral infection including that of *SARS-CoV-2* predisposes to secondary bacterial superinfection. We acknowledge that the enormous bacterial diversity observed in our study could potentially represent components of the normal microbiota. However, we would like to clarify that our hypothesis does not propose that superinfection is the sole factor driving bacterial diversity. Rather, we suggest that superinfection may contribute to increased diversity in certain cases. The severity of symptoms is influenced by various factors, including the host immune response, which can explain why some individuals with superinfection experience mild or no symptoms. While our manuscript does not provide specific information about the health status or infection sources of patients, we believe that further research is necessary to explore these aspects.

In healthy adults, skin-associated bacteria are usually present in the nasal cavity, predominantly representatives of *Actinobacteria* (e.g., *Corynebacterium* spp., *Propionibacterium* spp.), followed by *Firmicutes* (e.g., *Staphylococcus* spp.) and *Proteobacteria* (Bassis et al., 2014). In negative controls, we identified 58 species out of more than 900, thus contributing only about 6% to the microbial diversity and richness. The prevalent phyla were *Actinobacteria*, but the dominantly invaded species was *Stenotrophomonas maltophilia* belonging to *Proteobacteria* (*Pseudomonadota*). It has relatively low virulence in immunocompetent persons, but in patients with chronic respiratory disease, immunocompromization, prolonged antibiotic use (especially carbapenems) and long-term hospitalization or admission to the intensive care unit can be a source of life-threatening complications [38]. 

Not surprisingly, the nasopharyngeal virome in asymptomatic, mild, and severe patients was composed predominantly of different *SARS-CoV-2* variants, which correlated unambiguously with the period of occurrence of the particular strains in Slovakia. The sensitivity of metatranscriptome sequencing was only about 60%, but on the other hand, the set of *SARS-CoV-2* positive samples consisted of different Ct values. At the individual level, for all samples that had Ct of 20 or less, corresponding to higher viral loads, we were able to assemble the whole genome of the *SARS-CoV-2* virus. Samples from a group of severe patients were collected 3–7 days after inclusion in the study, and these patients had been treated in the interim with a wide range of drugs (antibiotics, corticotherapy, immunomodulatory therapy, antivirals) that may have influenced viral load in the meantime. The mild, asymptomatic, and healthy controls, based on questionnaire information, were not taking antimicrobial therapy or any other treatment with the potential to affect the microbiome. Despite the limited number of samples in the groups with confirmed *Delta*, *Alpha*, and *Omicron* variants, we were interested in whether they differed specifically at the level of bacterial taxa. The relative abundance of differentially repressed genera by the *SARS-CoV-2* WHO variant (LefSE) figure shows that the genus *Streptococcus* is clearly more abundant in the *Delta*-positive group compared to *Alpha* and *Omicron*. *Campylobacter* and *Haemophilus* are slightly enriched in the *Alpha* group (Figure 2E). The clear advantage of the transcriptome approach is its ability to capture not only RNA, but also DNA viruses (their transcripts). Besides *SARS-CoV-2*, unsurprisingly, the most frequently identified sequences were those derived from DNA viruses of bacteria, i.e., phages of the families *Pedoviridae*, *Rountreeviridae*, and *Siphoviridae*.

In one patient with severe COVID-19, we identified *Herpes Simplex Virus 1 (HSV-1)* transcripts corresponding to strain 17 (Genebank: JN555585.1) that mapped to the *LAT* (latency-associated transcript) region. *HSV-1* is a member of the *Herpesviridae* family and is known for establishing a latent infection in the sensory neurons. During the latent phase, the virus can persist in a dormant state within the host cells, primarily in the trigeminal ganglia [39]. The latency-associated transcripts (LATs) are a group of viral RNA molecules produced during this latent phase. These transcripts have been studied for their involvement in modulating the *HSV-1* latency and reactivation cycle. LATs have been implicated in various processes, such as promoting neuronal survival, reducing apoptosis, and modulating the immune response within the host [40]. However, it is important to note that the presence of *HSV-1* LATs alone may not be definitive evidence of active viral replication or reactivation. We can only speculate that immunosuppression caused by SARS Co-V2, in turn, could facilitate the reactivation of latent *HSV-1*.

We identified *Rhinovirus A* transcripts (Genebank: NP_042288) from the *Picornaviridae* family in two patients from the healthy control group. Rhinoviruses are a group of viruses that commonly cause respiratory infections, including the common cold. However, it is important to note that rhinoviruses typically cause mild respiratory symptoms, and many infections can go unnoticed or present with very mild symptoms. Rhinoviruses have been known to persistently shed in some individuals even in the absence of symptoms. In these cases, the virus can be intermittently detectable in respiratory samples, indicating a state of viral persistence without causing active infection or disease. Persistent shedding of rhinoviruses has been reported in both healthy individuals and those with respiratory conditions [41].

*Human Mastadenovirus D* (Genebank: NC_001617.1) from *Adenoviridae* family was found in one asymptomatic patient. The mapping transcript corresponds to the *E2B* gene, which is an early gene encoding the DNA polymerase. The virus belongs to the adenoviruses known to cause a variety of respiratory and ocular infections in humans. Adenovirus infections can range in severity from asymptomatic or mild illness to severe respiratory disease [42].

Using the Maaslin2 statistical tool, which allows analysis of abundance data along with age, sex, ATB, and SARS CoV-2 strain, demonstrated that age had no discernible effect on the abundance of bacterial transcripts in COVID-19 negative samples when a continuous scale for age was used. When employing a categorical scale for age, we observed significant effects on six taxa (Appendix A). In contrast, when considering all samples, we found that the 231 taxa exhibited variations associated with age (Appendix A). Notably, when focusing solely on severe patient samples (given that severe patients tended to be older than the other groups), we identified a distinct and pronounced influence of age on the data (Appendix A). Specifically, 114 taxa demonstrated significant associations with ATB treatment in severe patients (refer to Appendix A), despite a lack of significance observed in a prior simple *t*-test regarding the levels of the most abundant phyla (*p*-values displayed in Figure 2D). Furthermore, no statistically significant results were detected for the gender category in any of the aforementioned statistical analyses. These findings highlight the complexity of factors influencing the bacterial community composition in relation to age, with age exerting a more pronounced effect in severe patients. Further research is warranted to explore the underlying mechanisms and potential clinical implications of these age-related associations in bacterial abundance.

Sequences mapped to the human transcriptome were a non-negligible part of the analysis, but they are not the subject of this paper and deserve special attention.

## 5. Conclusions

The study analyzed the microbiome and virome of the nasopharynx in different groups of COVID-19 patients, including asymptomatic, mild, and severely affected individuals, as well as a control group. Metatranscriptome shotgun sequencing was used to analyze 151 samples. In the prokaryotic microbiome analysis, the relative abundance of bacteria was significantly higher in asymptomatic and mild patients compared to severe patients and the control group. *Firmicutes*, *Bacteroidetes*, *Proteobacteria*, *Actinobacteria*, and *Fusobacteriales* were the most abundant phyla in the positive groups, while *Actinobacteria*, *Proteobacteria*, and *Firmicutes* were dominant in the control group. The abundance of bacterial taxa was maintained even in severe patients who received antibiotic treatment. The use of antibiotics in severe patients decreased the overall bacterial abundance, but did not significantly alter the relative abundance of different bacterial phyla. Similar microbiome compositions were observed in both mild and asymptomatic groups. In virome analysis, SARS-CoV-2 genomes were assembled in 62% of the positive samples, with Alpha and Delta variants being predominant in severe and mild groups, respectively. Other human RNA viruses and phages from various families were also detected in the samples. Overall, this study provides insights into the composition and diversity of the microbiome and virome in COVID-19 patients, highlighting differences based on disease severity and the presence of SARS-CoV-2 variants.

## Figures and Tables

**Figure 1 microorganisms-11-01804-f001:**
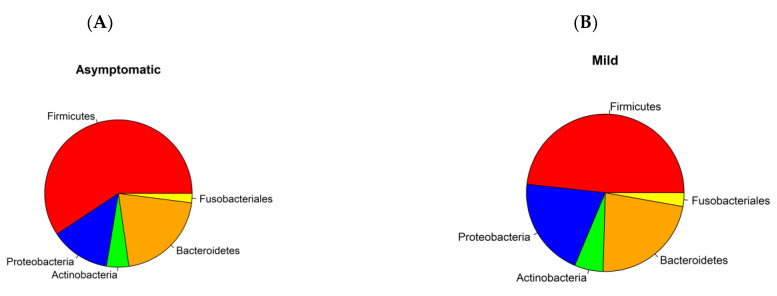
Pie charts illustrating the distribution of 5 most frequent taxa in asymptomatic (**A**), mild (**B**), severe (**C**), and negative control groups (**D**).

**Figure 2 microorganisms-11-01804-f002:**
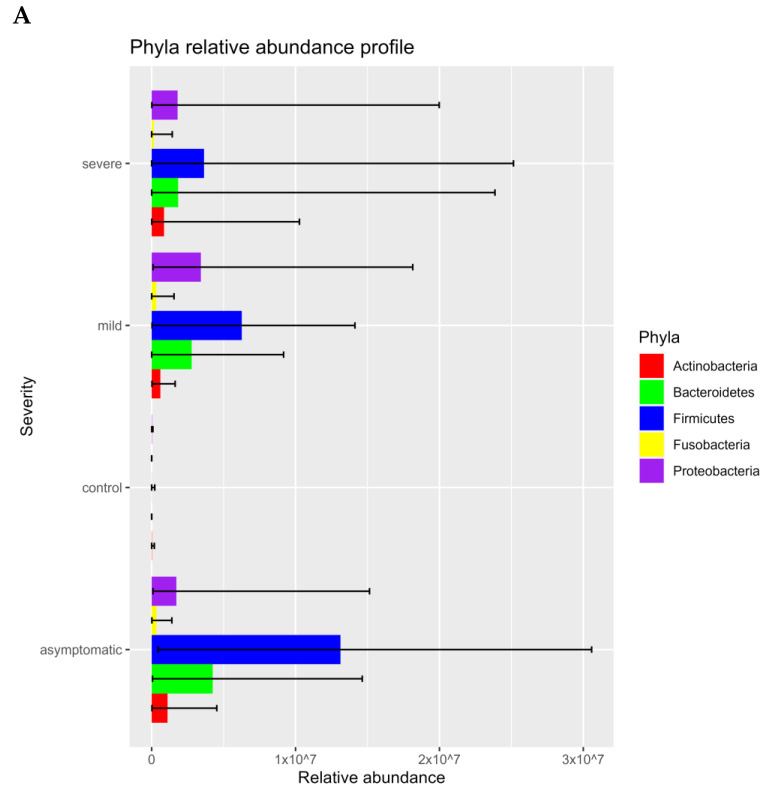
Horizontal bar charts illustrating the relative abundance of phyla (**A**) according to the severity of COVID-19 (**B**), phyla abundance according to ATB treatment in severe patients (**C**), statistical comparison of relative phyla abundance in ATB and non- ATB group (only severe) (**D**), and relative abundance of differentially represented genera by WHO SARS-CoV-2 variants, shown genera were chosen under conditions: LDA from LefSE > 4 and median of relative abundance in either control group or COVID-positive group > 100 (**E**).

**Figure 3 microorganisms-11-01804-f003:**
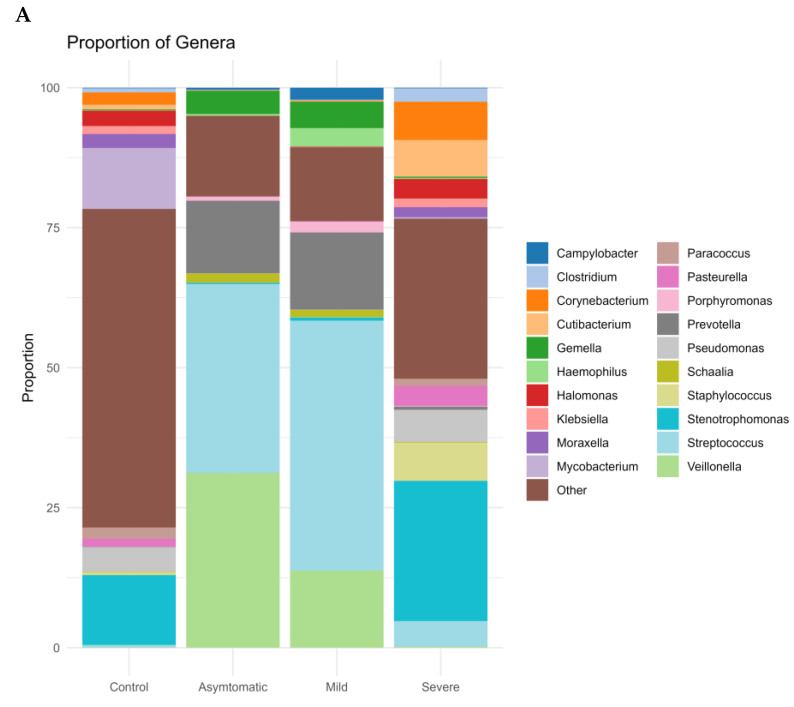
Stacked bar graphs illustrating relative abundance in all groups of (**A**) bacterial genera and (**B**) bacterial species.

**Table 1 microorganisms-11-01804-t001:** Study cohort, Ct mean values (only severe, mild, asymptomatic).

	Number	Median (Years)
**Age**	151	39.5 (17–90)
**Negative controls**	72	37 (25–75)
male	26	42 (25–58)
female	46	35 (25–75)
**Severe**	30	68 (32–90)
male	16	69.5 (32–90)
female	14	64 (41–77)
**Mild**	25	37 (17–57)
male	13	38 (17–57)
female	12	32.5 (19–57)
**Asymtomatic**	24	42 (20–49)
male	14	39 (21–49)
female	10	43 (36–48)
**BMI**	132	
**Negative controls**	57	24.48 (17.2–42.4)
male	25	28.4 (19.7–42.4)
female	32	21 (17.2–32.6)
**Severe**	19	30.39 (23.33–49.21)
male	8	30 (23.66–34.60)
female	11	30 (23.33–49.21)
**Mild**	25	25.64 (18.62–33.46)
male	13	28.27 (21.32–33.46)
female	12	21.71 (18.62–29.15)
**Asymtomatic**	24	25.72 (16.9–32.5)
male	14	28.10 (23.54–32.50)
female	10	20.73 (16.9–30.11)
**Associated diseases**	38/151	25.16%
Diabetes	12/151	7.94%
severe	10	33.33%
mild	1	4%
negative controls	1	1.30%
**Arterial hypertension**	21/151	13.90%
severe	15	50%
asymtomatic	3	12.50%
negative controls	2	2.70%
mild	1	4%
Ischaemic heart disease	7/151	4.63%
only severe	7	23.33%
**Oncological disease**	4/151	2.64%
severe	3	10%
negative controls	1	1.30%
**Multiple diagnosis (more than 2 )**	11/151	7.28%
severe	10	33.33%
mild	1	4%
**ATB (severe only)**	30	
Ruzinov hospital	18	
ATB	18	100%
no ATB	0	0%
Kramare hospital	12	
ATB	2	16.70%
no ATB	10	83.30%

	**Median Ct value E gene**	**SD**
**Severe**	28.06 (13.9-37.3)	7.34
**Mild**	24.89 (15-38.77)	5.32
**Asymptomatic**	28.56 (19.24-33.59)	4.72

SD—standard deviation, Ct—threshold cycle in RT-PCR.

## Data Availability

Data from metatranscriptome sequencing are available on request.

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
