# Peer review of "Metatranscriptome Analysis of Nasopharyngeal Swabs across the Varying Severity of COVID-19 Disease Demonstrated Unprecedented Species Diversity"

_microorganisms, 2023, doi:10.3390/microorganisms11071804_

Round 1

Reviewer 1 Report

The manuscript presents an interesting study on the microbial community differences among various COVID-19 severity groups, with a robust bioinformatics pipeline. However, the statistical analysis would benefit from the use of a Generalized Linear Model to better account for confounding factors.

Major comment:

1. I suggest using a Generalized Linear Model (GLM) to analyze the abundance data, accounting for age, sex, and antibiotic treatment. Then, perform an analysis of deviance test to assess group differences' significance. Examine the GLM output's coefficients to identify key genera linked to each group while considering confounding factors. Visualize the relationships using adjusted plots and account for multiple comparisons with methods like Bonferroni, Holm, or false discovery rate (FDR).

Minor Comments:

1. Improve the quality of figures.

2. Replace pie charts with bar plots sorted by abundance.

3. Remove the gray background from some plots.

4. Figure 2: Focus on the top genera by showing only the top 10-20 most abundant or most relevant genera across the groups, which will reduce visual clutter and allow readers to focus on key taxa. Group less abundant or less relevant genera into a single "Other" category to simplify the bar plot without losing important information. Use a distinguishable color palette, and increase the font size for better readability.

5. The conclusion appears unfocused and could be improved for better clarity. Please provide a clearer summary of the main findings and their implications.

Author Response

I truly appreciate your feedback, and I'm grateful for your valuable input. I hope I have been able to meet your major and minor requests for improvement. Please see attached file.

Reviewer 2 Report

In this manuscript, the authors used metatranscriptomics to assess the microbiome of people infected with SARS-CoV-2. The authors included different groups of patients in the analysis, including uninfected, asymptomatic, mildly and severely ill patients, totaling 151 people. Using high-throughput sequencing, the authors observed a significant increase in microbial species diversity in patients with COVID-19, regardless of disease severity. Furthermore, they describe a greater relative abundance of bacteria in the mild and asymptomatic groups and a small proportion of bacteria in the severe and uninfected groups.

The manuscript is interesting and the methods are suitable for the main purpose of the study. Despite the enormous amount of data, the authors failed to explore the results, basically indicating the diversity of microorganisms, but without major biological inferences. Dozens of SARS-CoV-2 genomes have been fully assembled, but there is no analysis on this. The figures are of poor quality and the discussion needs to be deeply improved. More importantly, there are other studies focusing on metatranscriptome analysis of patients in the context of COVID-19 (eg doi 10.3389/fcimb.2022.1011672 and 10.3390/biom13010006), which reduces the impact of the present study. Given these circumstances, I feel this manuscript is not suitable for publication in this journal.

However, I have included some points that the authors could work on and try to improve the quality of the manuscript for future submission.

Major points:

1) The authors found less bacterial diversity in non-infected patients (control), but the amount of reads is very different from the others (only 0.1% of total reads). How to be sure that this result is true or not just a bias due to lack of data?

2) The authors argue that asymptomatic people and patients with mild disease had greater bacterial diversity due to a possible situation of bacterial superinfection. However, there is an enormous bacterial diversity, most likely being components of the normal microbiota. If superinfection is true, how come these patients had only mild symptoms or no symptoms at all? And how would these patients get infected? Do the authors have information about the health status of these patients? It is a very curious hypothesis that they bring to the discussion, but it lacks more solid evidence.

3) What was the rationale for considering 1000 reads for species demarcation? Is there a precedent in the literature? If so, this must be clearly indicated.

4) The figures are of very low quality, with letters too small to be read. In addition, there is a lot of information in the figures, especially in figure 1, which makes it difficult to fully understand the results. Perhaps the authors could remake the figure and try to improve the quality.

Minor points:

1) Taxon names should be revised in italics. Sometimes names are italicized, sometimes not. This must be fixed considering the current taxonomy rules;

2) Table 1: Some numbers are in red apparently for no reason. This must be fixed;

3) Line 211: replace 'flora' with 'microbiota';

4) Line 235: Check the numbers here. Are there 41 or 51 genomes?

5) Lines 247-249: What can you discuss about finding RNA in these samples belonging to DNA viruses?

Author Response

I truly appreciate your feedback, and I'm grateful for your valuable input. I hope I have been able to meet your major and minor requests for improvement. Please see the attached file.

Reviewer 3 Report

Summary:

The authors of the study attempt to characterize the airway microbiome of COVID-19 patients with different levels of severity. A metatranscriptomic approach is employed. Results show a higher diversity in COVID-19 patients when compared to controls. Additionally, COVID-19 variant-specific changes are observed.

Major comments:

-       The use of metatranscriptomic should be questioned. Indeed, it does allow the identification of both RNA and DNA viruses however, the analysis of viral data is barely touched upon across the manuscript, and given the use of antibiotics, this type of data introduces a non-neglectable bias.

-       Given the available descriptions, it is unclear which criteria were used to select controls. Were they healthy negative controls? Symptomatic negative controls? It is essential that more details are provided, as well as a comparison of demographics for those controls.

-       The authors report that no association with severity was highlighted however, given the use of antibiotics, one could argue that this question cannot be answered from the available data.

-       The fact that severe patients were recruited from two different hospitals, while others were recruited from a different structure is quite concerning. The current design makes it impossible to assess if the observations reported here are specific to the question of interest, or the structures where the patients were recruited. If possible, a sensitivity analysis only including non-severe patients, as well as an additional analysis only including severe patients and comparing both hospitals, should be presented. Unless this is provided, no robust conclusions can be derived from the presented work.

-       It is absolutely essential, for this type of study, that demographics are presented, as well as labs, if available, and outcomes. The relevance of splitting the main table by gender is unclear, and the authors might consider presenting those as sub-rows, rather than different columns that make the table difficult to read. The demographics table should also present a breakdown by structure/hospital that could be added as supplementary.

-       Figures 1A and B are irrelevant. Indeed, the distributions are almost entirely explained by the use of antibiotics in severe patients. Given the use of antibiotics in some of the patients, presenting absolute measures must be avoided.

-       There is no mention of how the data was processed to deal with contaminants and commensal organisms. Contamination is a big concern in this type of analysis and should be at least discussed.

-       The impact of the work is unclear. I would advise the authors to work on presenting the results so that this point is clear.

Minor comments:

-       The current title is very generic and does not mention COVID-19 at all, despite the study being specific to COVID-19.

-       Some terms and phrasing used are imprecise and hinder the credibility and impact of the work presented:

o   Line 3 – The term “multifocus” is used, but its meaning is unclear.

o   Lines 16-18 – It appears that Actinobacteria and Bacteroides were less abundant in COVID-19 patients. The current phrasing can be misleading and I would advise rephrasing this section.

o   Lines 24-25 – The use of the word “certain” is not specific enough and a description is required.

-       Lines 18-20 – “the same” might have been mistakenly added to that sentence.

-       Line 30 – The section “perhaps as many as a million a day” is not relevant and completely speculative unless a reference is provided.

-       The number of patients included needs to be included in the abstract.

-       Figures 1D and 1E are referred to after 1H. This needs to be reordered.

-       The Venn diagrams present too many comparisons and are thus, impossible to read. I suggest the authors remove those and consider alternative representations.

-       Figure 2 is very crowded. The authors might consider grouping using higher taxonomic groups.

-       Figure 1J is referred to after figure 2 and should be reordered accordingly. It is essential that a legend is added as well. A figure presenting odd ratios or a volcano-like figure might be easier to interpret.

-       An analysis comparing antibiotics patients vs. non-antibiotics patients is presented in the discussion. No new results should be presented in the discussion. This should appear in the results section.

acceptable

Author Response

(The authors gave the same response as above.)

Round 2

Reviewer 2 Report

The authors responded adequately to all the points I raised. The manuscript has been improved with new data interpretation and the results are now more robust. Also, the quality of figures has been greatly improved. Congratulations to the authors for the good work during this review process.

Author Response

Once again, I would like to express my gratitude for your valuable comments, which will improve the quality of the work presented. I hope that this time it will be sufficient.

Reviewer 3 Report

I thank the authors for carefully revising their manuscript. It is much improved, and I only have a handful of comments.

Table 1 is still not complete enough. For studies such as this one, more data is required for interpretation. As mentioned previously, I believe race, BMI, survival, a metric/indicator of respiratory function, and death, split by study site must be reported. Additionally, the presentation of the current table could be improved.

To follow up on the commensal/contaminant issue, water controls are sometimes used to deal with background-related processing. Bioinformatic algorithms are currently not well-powered to do this accurately. I think this should be commented on.

I agree that Venn diagrams are purely illustrative, but there are too many comparisons and thus the graphs are not readable. The impact of your findings would only be maximized by clearer visuals.

/

Author Response

(The authors gave the same response as above.)
